# Conserved momenta of ferromagnetic solitons through the prism of differential geometry

Xingjian Di[1⋆] and Oleg Tchernyshyov[2†]

**1** Department of Physics, University of Illinois at Urbana–Champaign, Urbana, IL 61801, USA
**2** Department of Physics and Astronomy, Johns Hopkins University, Baltimore, MD 21218, USA

† olegt@jhu.edu
⋆ Currently at: Courant Institute of Mathematical Sciences, New York University, New York, NY 10012, USA.

## Abstract

The relation between symmetries and conservation laws for solitons in a ferromagnet is complicated by the presence of gyroscopic (precessional) forces, whose description in the Lagrangian framework involves a background gauge field. This makes canonical momenta gauge-dependent and requires a careful application of Noether's theorem. We show that Cartan's theory of differential forms is a natural language for this task. We use it to derive conserved momenta of the Belavin–Polyakov skyrmion, whose symmetries include translation, global spin rotation, and dilation.

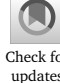

# 1 Introduction

## 1.1 Solitons and collective coordinates

Magnetic systems are known to have a variety of soliton solutions that are stable for dynamical or topological reasons [1]. Of particular interest are topological solitons—e.g., domain walls, vortices and skyrmions—whose stability is ensured by their nontrivial topology. Although a magnetization field has an infinite number of degrees of freedom, the low-energy physics of a soliton can usually be reduced to a few soft modes parametrized by collective coordinates. A classic example of this collective approach is the work of Thiele [2], who derived the equations of motion for the position $\mathbf{R}(t)$ of a rigidly moving magnetic soliton:

$$-\frac{\partial U}{\partial \mathbf{R}} + \mathbf{G} \times \dot{\mathbf{R}} = 0, \tag{1}$$

where the dot represents the time derivative. Eq. (1) expresses a dynamical balance of forces acting on the soliton. The first term represents a conservative force arising from the dependence of the potential energy $U$ on the soliton position. The second, gyroscopic force is the analog of the Lorentz force acting on a moving electric charge in a background magnetic field $\mathbf{G}$. The role of the magnetic field here is played by the gyrovector $\mathbf{G}$. We omitted the force of viscous friction $-D\dot{\mathbf{R}}$ in Eq. (1), which spoils conservation laws.

Thiele's approach was extended [3, 4] to an arbitrary set of collective coordinates $q \equiv \{q^1, q^2, \ldots, q^N\}$. These equations of motion also express the balance of (generalized) forces for each collective coordinate $q^i$:

$$-\partial_i U + F_{ij}\dot{q}^j = 0. \tag{2}$$

Here $\partial_i$ is a shorthand for $\partial/\partial q^i$; Einstein's summation convention is assumed. The antisymmetric gyroscopic tensor $F_{ij}$ is the inverse of the Poisson bracket [5].

## 1.2 Lagrangian of a ferromagnetic soliton

The equations of motion (2) can be derived from the following Lagrangian reminiscent of a massless electrically charged particle in a background magnetic field [6]:

$$L = A_i(q)\dot{q}^i - U(q). \tag{3}$$

The curl of the gauge potential $A_i$ gives a magnetic field equal to the gyroscopic tensor,

$$F_{ij} = \partial_i A_j - \partial_j A_i. \tag{4}$$

As in electromagnetism, the gauge potential $A_i$ is not a physical quantity as it can be changed without affecting the equations of motion. A gauge transformation

$$A_i \mapsto A'_i = A_i + \partial_i \lambda, \tag{5}$$

where $\lambda(q)$ is an arbitrary scalar function, leaves the gyrosocopic tensor $F_{ij}$ invariant.

## 1.3 Canonical and conserved momenta

When a system exhibits a continuous symmetry, Noether's theorem guarantees the existence of a conserved quantity. Say, both the potential energy and the gyroscopic tensor are independent of a particular coordinate $q^a$,

$$\partial_a U = 0, \quad \partial_a F_{ij} = 0. \tag{6}$$

We then naturally expect that there is a conserved momentum. A naive application of Noether's theorem suggests that the conserved quantity is the canonical momentum

$$p_a = \frac{\partial L}{\partial \dot{q}^a} = A_a. \tag{7}$$

However, this educated guess runs into a potential problem: this momentum is gauge-dependent and is therefore unphysical. The formal reason behind this problem can be traced to the ambiguity of the gauge potential: even though the "magnetic field" $F_{ij}$ is symmetric under a $q^a$ translation, the gauge potential $A_i$ may not be. As Weinberg [7] put it, the Lagrangian is not symmetric, only the action is. A careful application of Noether's theorem yields the following result for the conserved momentum [5]:

$$P_a(2) - P_a(1) = -\int_1^2 F_{ai}(q)\,dq^i. \tag{8}$$

Here the integration path follows a line connecting points $q = 1$ and $q = 2$ in the space of collective coordinates. Ref. [5] examines in detail conserved momenta of a domain wall in 1 dimension and of a vortex in 2 dimensions.

## 1.4 Outstanding questions

The result for the conserved momentum (8) appears unfamiliar and counterintuitive. It raises immediate questions.

For example, what is the relation between canonical (7) and conserved (8) momenta? Can the former be used in place of the latter? The answer is sometimes yes, and sometimes no. In a fixed gauge, only some of the conserved momenta may coincide with their canonical counterparts, but not all [5]. If we choose a gauge invariant under $q^a$ translations, $\partial_a A_i = 0$, then $F_{ai} = -\partial_i A_a$ and Eq. (8) yields $P_a = A_a = p_a$. It is generally not possible to choose a gauge

invariant under all symmetries [5], so one needs to use a different gauge for each conserved momentum if one wishes to use the canonical route.

Furthermore, does the value of the conserved momentum change if we follow a different path from 1 to 2? The short answer is that the result is not sensitive to the path's geometry but may depend on its topology. It can be shown [5] that an infinitesimal deformation of the path does not change the answer. The proof relies on the Bianchi identity for the gyroscopic tensor,

$$\partial_i F_{jk} + \partial_j F_{ki} + \partial_k F_{ij} = 0 \,. \tag{9}$$

By extension, any two homotopic[1] paths with the same endpoints will yield the same conserved momentum (8). However, two non-homotopic paths may give different answers for the conserved momentum [8].

## 1.5 Structure of the paper

In this paper we reexamine the relation between symmetries and conserved momenta of a ferromagnetic soliton from a geometric perspective. Cartan's theory of differential forms provides a natural language to link symmetries with conserved momenta in this context. The analogy between the gyroscopic tensor and magnetic field mentioned above helps make a case for the use of differential forms. Although vector calculus remains the standard mathematical language in which physics students learn Maxwell's electrodynamics [9], theory of differential forms provides a more concise, insightful and versatile perspective [10–12]. The language of differential forms is naturally suited for working with smooth manifolds beyond Euclidean spaces and is thus particularly convenient for a space of arbitrary collective coordinates.

In the remainder of the paper we first discuss the link between symmetries and conserved momenta in the language of differential geometry in Section 2. Section 3 illustrates these general considerations on the well-studied examples of a domain wall in 1 dimension and of a vortex in 2 dimensions. Conserved momenta of the Belavin–Polyakov skyrmion [13], associated with the symmetries of translation, global spin rotation, and dilation, are derived in Section 4. Concluding remarks are given in Section 5.

# 2 Geometrical formulation

## 2.1 Mathematical preliminaries

We will work with a differentiable manifold $\mathcal{M}$ parameterized by coordinates $q = \{q^1, \dots, q^N\}$. A differential $n$-form $\omega$ with a vanishing exterior derivative, $d\omega = 0$, is said to be *closed*. It is called *exact* if it can be expressed as an exterior derivative of an $(n-1)$-form, $\omega = d\eta$. Form $\eta$ is known as the *potential* of form $\omega$ [14].

An exact form is always closed: $d\omega = dd\eta = 0$. The converse is not necessarily true. A potential $\eta$ for a closed form $\omega$ can always be constructed locally. However, the global definition of $\eta$ on the entire manifold $\mathcal{M}$ is not always possible [10, 14]. De Rham's theorem establishes a necessary and sufficient condition under which a closed form has a globally defined potential (Appendix A).

The gyroscopic tensor $F_{ij}$ gives rise to a 2-form

$$F = \frac{1}{2} F_{ij} \, dq^i \wedge dq^j \,, \tag{10}$$

---

[1]Two curves are homotopic if one can be continuously deformed into another.

known as the symplectic form, the inverse of the Poisson tensor [14]. The Bianchi identity (9) indicates that the symplectic form is closed,

$$dF = 0 \,. \tag{11}$$

Therefore, we may construct the potential of $F$, which is at least locally defined, so that

$$F = dA \,. \tag{12}$$

The potential 1-form $A$ is related to the gauge potential,

$$A = A_i \, dq^i \,. \tag{13}$$

Continuous symmetries are associated with vector fields

$$V = V^i \partial_i \,. \tag{14}$$

For example, $\partial_a$ is the generator of infinitesimal translations for coordinate $q^a$. An infinitesimal rotation in the $(q^a, q^b)$ plane is generated by the vector field $q^a \partial_b - q^b \partial_a$.

The operation of interior product (contraction) $\iota$ takes a vector field $V$ and an $n$-form $\omega$ and yields an $(n-1)$ form $\iota_V \omega$ [14]. For the 1-form $A$ and the 2-form $F$,

$$\iota_V A = V^i A_i \,, \quad \iota_V F = V^i F_{ij} \, dq^j \,. \tag{15}$$

The interior product allows us to write the equation of motion (2) in a coordinate-free form:

$$\iota_D F = -dU \,. \tag{16}$$

Here

$$D = \frac{d}{dt} = \dot{q}^i \partial_i \tag{17}$$

is the vector field expressing time evolution in the space of collective coordinates.

Another useful mathematical concept is the Lie derivative $\mathcal{L}_V$ for a vector field $V = V^i \partial_i$. For an $n$-form $\omega$, the Lie derivative $\mathcal{L}_V \omega$ is also an $n$-form. For a 0-form (function) $f$, the Lie derivative is simply the directional derivative:

$$\mathcal{L}_V f = V^i \partial_i f = \iota_V df \,. \tag{18}$$

The Lie derivative of a basis 1-form $dq^i$ is

$$\mathcal{L}_V dq^i = \partial_j V^i \, dq^j = d(\iota_V dq^i) \,. \tag{19}$$

Eqs. (18) and (19) together with the Leibniz differentiation rule allow one to compute the Lie derivative of any $n$-form. E.g., for a 1-form $A = A_i \, dq^i$ we obtain

$$\mathcal{L}_V A = \mathcal{L}_V (A_i \, dq^i) = (\mathcal{L}_V A_i) \, dq^i + A_i \, \mathcal{L}_V dq^i = V^j \partial_j A_i \, dq^i + A_i \partial_j V^i \, dq^j \,. \tag{20}$$

Of particular importance for us will be Cartan's magic formula [14]

$$\mathcal{L}_V \omega = d(\iota_V \omega) + \iota_V d\omega \,. \tag{21}$$

## 2.2 Cartan symmetries and conserved momenta

The invariance of the potential energy $U$ and of the symplectic form $F$ under a symmetry generated by a vector field $V$ is expressed as the vanishing of their Lie derivatives:

$$\mathcal{L}_V U = 0, \quad \mathcal{L}_V F = 0. \tag{22}$$

A symmetry preserving both the potential energy $U$ and the symplectic form $F$ is known as a Cartan symmetry [14].

For each Cartan symmetry $V$ we can construct a conserved momentum $P_V$ as follows. Observe that the 1-form $\iota_V F$ is closed:

$$d(\iota_V F) = \mathcal{L}_V F - \iota_V dF = 0. \tag{23}$$

Here we used Cartan's formula (21), the invariance of the symplectic form (22), and its closed nature (11). The closed 1-form $\iota_V F$ can then be expressed in terms of a potential, a 0-form (function) $P_V$:

$$\iota_V F = -dP_V. \tag{24}$$

As mentioned in Sec. 2.1, the potential $P_V$ is defined locally but not necessarily globally.

The function $P_V$ can now be obtained by integrating the 1-form $\iota_V F$ along a line $C$:

$$\int_{\partial C} P_V = \int_C dP_V = -\int_C \iota_V F. \tag{25}$$

Here $\partial C$ is the (oriented) boundary of line $C$ consisting of its endpoints 1 and 2. Returning to the coordinate notation, we obtain

$$P_V(2) - P_V(1) = -\int_1^2 V^i(q) F_{ij}(q) \, dq^j. \tag{26}$$

For $V = \partial_a$ (translations of coordinate $q^a$), this result reproduces Eq. (8).

Having defined $P_V$, let us now show that it is a constant of motion. With the aid of the equation of motion (16) and of the invariance of potential energy (22), we obtain

$$\dot{P}_V = \iota_D dP_V = -\iota_D \iota_V F = \iota_V \iota_D F = -\iota_V dU = -\mathcal{L}_V U = 0. \tag{27}$$

We thus find that $P_V$ remains constant in time.

For translations, $V = \partial_a$, the rate of change

$$\dot{P}_a = -\iota_{\partial_a} dU = -\partial_a U \tag{28}$$

is equal to the force conjugate to $q^a$, so it makes sense to regard $P_a$ as the momentum for coordinate $q^a$.

## 2.3 Is a conserved momentum single-valued?

The definition of conserved momentum $P_V$ is based on the converse of Cartan's lemma [10,14]: locally, a closed 1-form $\iota_V F$ is also exact,

$$d(\iota_V F) = 0, \quad \text{therefore} \quad \iota_V F = -dP_V. \tag{29}$$

However, the resulting 0-form $P_V$ is only guaranteed to be defined in a local coordinate chart that is simply connected. In other words, the function $P_V(q)$ may not be well-defined on the entire manifold of collective coordinates.

We can see how an attempt to define potential $P_V$ globally can fail. Eq. (25) gives the potential increment between the endpoints of line $C$. This increment should vanish if the line is a loop. So the 1-form $\iota_V F$ must satisfy the condition

$$\int_C \iota_V F = 0 \,, \tag{30}$$

for any loop $C$ in the manifold of collective coordinates. For a closed 1-form, this integral turns out to be a topological quantity: a continuous deformation of the loop does not change it (see Appendix A). Therefore, if the manifold of collective coordinates is simply connected —in the sense that all loops are contractible to a point—then all such integrals vanish and the potential $P_V$ is globally defined. On the other hand, in a manifold with non-contractible loops may have a non-zero $\int_C \iota_V F$. Then the potential $P_V$ becomes multivalued. See Appendix A for more details and Section 3.1 for an example of a multi-valued conserved momentum.

### 2.4 Conserved momenta from canonical momenta

We mentioned in the introduction that a conserved momentum $P_a$ (8) coincides with the canonical momentum $p_a$ (7) in cases when the vector potential $A_i$ is symmetric under translations of coordinate $q^a$ with the generator $\partial_a$. This argument applies more broadly to any Cartan symmetry generated by a vector field $V = V^i \partial_i$.

Begin with Eq. (24) that defines conserved momentum $P_V$ in terms of the symplectic 2-form $F = dA$ and use Cartan's formula (21):

$$-dP_V = \iota_V F = \iota_V dA = \mathcal{L}_V A - d\,\iota_V A. \tag{31}$$

Choose a gauge 1-form $A$ that is invariant under the Cartan symmetry $V$,

$$\mathcal{L}_V A = 0\,. \tag{32}$$

It then follows that

$$P_V = \iota_V A = V^i A_i = V^i p_i\,. \tag{33}$$

Note that the choice of a symmetric gauge is not unique. Any other gauge

$$A' = A + d\lambda\,, \tag{34}$$

where $\lambda$ is a symmetric function, $\mathcal{L}_V \lambda = 0$, is also symmetric. This creates no ambiguity for the conserved momentum:

$$P_V' = \iota_V A' = \iota_V A + \iota_V d\lambda = P_V + \mathcal{L}_V \lambda = P_V\,. \tag{35}$$

### 2.5 Single classical spin

A classical spin is described by a 3-component vector of a fixed length $S$ and can be parametrized, e.g., by the polar angle $\theta$ and the azimuthal angle $\phi$:

$$\mathbf{S} = (S^1, S^2, S^3) = S(\sin\theta\cos\phi, \sin\theta\sin\phi, \cos\theta)\,. \tag{36}$$

The equation of motion describes precession about the effective field $-\partial U/\partial \mathbf{S}$:

$$\dot{\mathbf{S}} = -\mathbf{S} \times \frac{\partial U}{\partial \mathbf{S}}\,. \tag{37}$$

Expressed in the angular coordinates, it reads

$$-S \sin\theta \, \dot\phi = -\frac{\partial U}{\partial \theta}, \quad S \dot\theta = -\frac{1}{\sin\theta}\frac{\partial U}{\partial \phi}. \tag{38}$$

These equations are equivalent to Eq. (2) with $F_{\theta\phi} = -F_{\phi\theta} = -S \sin\theta$.

The symplectic form

$$F = -S \sin\theta \, d\theta \wedge d\phi \tag{39}$$

is trivially closed (there are no 3-forms on a 2-dimensional manifold), $dF = 0$. It is proportional to the area element on the unit sphere $\sin\theta \, d\theta \wedge d\phi$ and is therefore invariant under rotations, $\mathcal{L}_V F = 0$, for the vector fields

$$\begin{aligned}
V_1 &= S^2 \frac{\partial}{\partial S^3} - S^3 \frac{\partial}{\partial S^2} = -\sin\phi \frac{\partial}{\partial \theta} - \cot\theta \cos\phi \frac{\partial}{\partial \phi}, \\
V_2 &= S^3 \frac{\partial}{\partial S^1} - S^1 \frac{\partial}{\partial S^3} = \cos\phi \frac{\partial}{\partial \theta} - \cot\theta \sin\phi \frac{\partial}{\partial \phi}, \\
V_3 &= S^1 \frac{\partial}{\partial S^2} - S^2 \frac{\partial}{\partial S^1} = \frac{\partial}{\partial \phi},
\end{aligned} \tag{40}$$

representing infinitesmial rotations about the three Cartesian axes. From these vector fields and the symplectic 2-form we obtain three closed, and therefore exact, 1-forms

$$\begin{aligned}
\iota_{V_1} F &= -S \cos\theta \cos\phi \, d\theta + S \sin\theta \sin\phi \, d\phi = -d(S \sin\theta \cos\phi) = -dS^1, \\
\iota_{V_2} F &= -S \cos\theta \sin\phi \, d\theta - S \cos\theta \sin\phi \, d\phi = -d(S \sin\theta \sin\phi) = -dS^2, \\
\iota_{V_3} F &= S \sin\theta \, d\theta = -d(S \cos\theta) = -dS^3.
\end{aligned} \tag{41}$$

As one might expect, the relevant momenta are the Cartesian components of spin $S^1$, $S^2$, and $S^3$. They are conserved if potential energy $U$ is invariant under rotations about the corresponding axes.

## 2.6 Many spins

For an ensemble of spins $\{\mathbf{S}_1, \ldots, \mathbf{S}_\mathcal{N}\}$, the equations of motion are similar to those for a single spin (38):

$$-S \sin\theta_\nu \, \dot\phi_\nu = -\frac{\partial U}{\partial \theta_\nu}, \quad S \dot\theta_\nu = -\frac{1}{\sin\theta_\nu}\frac{\partial U}{\partial \phi_\nu}, \quad \nu = 1, \ldots, \mathcal{N}. \tag{42}$$

The gyroscopic tensor is block-diagonal, with nonzero entries $F_{\theta_\nu \phi_\nu} = -F_{\phi_\nu \theta_\nu} = -S \sin\theta_\nu$. That yields the symplectic form

$$F = -S \sum_{\nu=1}^{\mathcal{N}} \sin\theta_\nu \, d\theta_\nu \wedge d\phi_\nu. \tag{43}$$

Like its single-spin counterpart (39), this 2-form is closed, $dF = 0$.

The energy of Heisenberg exchange interaction,

$$U = -\frac{1}{2} \sum_{\mu=1}^{\mathcal{N}} \sum_{\nu=1}^{\mathcal{N}} J_{\mu\nu} \mathbf{S}_\mu \cdot \mathbf{S}_\nu, \tag{44}$$

is invariant under global spin rotations. In infinitesimal form, these symmetries are represented by vector fields

$$V_1 = \sum_{\nu=1}^{\mathcal{N}} \left( S_\nu^2 \frac{\partial}{\partial S_\nu^3} - S_\nu^3 \frac{\partial}{\partial S_\nu^2} \right) = \sum_{\nu=1}^{\mathcal{N}} \left( -\sin\phi_\nu \frac{\partial}{\partial \theta_\nu} - \cot\theta_\nu \cos\phi_\nu \frac{\partial}{\partial \phi_\nu} \right) \tag{45}$$

and so on—cf. Eq. (41). They give rise to closed 1-forms

$$\iota_{V_1} F = \sum_{\nu=1}^{\mathcal{N}} (-S \cos \theta_\nu \cos \phi_\nu \, d\theta_\nu + S \sin \theta_\nu \sin \phi_\nu \, d\phi_\nu) = -d \sum_{\nu=1}^{\mathcal{N}} S_\nu^1 = -dS^1, \qquad (46)$$

etc., so the relevant conserved quantities are Cartesian components of the total spin $\mathbf{S} = \sum_{\nu=1}^{\mathcal{N}} \mathbf{S}_\nu$.

## 2.7 Collective coordinates

The state of spins $\{\mathbf{S}_1, \ldots, \mathbf{S}_{\mathcal{N}}\}$ can be parametrized in terms of collective variables $q \equiv \{q^1, \ldots, q^N\}$. The number of spins $\mathcal{N}$ is typically macroscopically large, whereas the number of collective coordinates $N$ is kept small to make such a parametrization practical. This map from the $N$-dimensional manifold of collective coordinates to a $2\mathcal{N}$-dimensional manifold of spin states allows us to define the symplectic form for the collective coordinates via pullback,

$$F = \frac{1}{2} F_{ij} \, dq^i \wedge dq^j, \qquad (47)$$

where the coefficients of the gyroscopic tensor for collective coordinates are

$$F_{ij} = -S \sum_{\nu=1}^{\mathcal{N}} \sin \theta_\nu \left( \frac{\partial \theta_\nu}{\partial q^i} \frac{\partial \phi_\nu}{\partial q^j} - \frac{\partial \theta_\nu}{\partial q^j} \frac{\partial \phi_\nu}{\partial q^i} \right) = -S \sum_{\nu=1}^{\mathcal{N}} \mathbf{m}_\nu \cdot \left( \frac{\partial \mathbf{m}_\nu}{\partial q^i} \times \frac{\partial \mathbf{m}_\nu}{\partial q^j} \right), \qquad (48)$$

where $\mathbf{m}_\nu = \mathbf{S}_\nu / S$ is the unit vector parallel to spin $\mathbf{S}_\nu$. As a pullback of a closed form (43), the symplectic form for collective coordinates (47) is also closed. That can also be checked directly, by verifying that the matrix elements of the gyroscopic tensor (48) satisfy the Bianchi identity (9).

# 3 Illustrative examples

## 3.1 Domain wall in a ferromagnetic wire

Consider the well-known example of a domain wall in a ferromagnetic wire. The energy functional includes Heisenberg exchange and easy-axis anisotropy. In natural units of energy and length,

$$U[\mathbf{m}(x)] = \int_{-\infty}^{\infty} dx \, \frac{1}{2} \left[ \partial_x \mathbf{m} \cdot \partial_x \mathbf{m} - (\mathbf{m} \cdot \mathbf{e}_3)^2 \right]. \qquad (49)$$

Here the unit vector $\mathbf{e}_3$ specifies the easy direction in spin space. The energy possesses the symmetries of translation and of global spin rotation about the easy axis defined by the unit vector $\mathbf{e}_3 = (0, 0, 1)$. The two ground states, minimizing the energy, are uniform, $\mathbf{m}(x) = \pm \mathbf{e}_3$.

The energy also has local energy minima in the form of domain walls,

$$m^1 + im^2 = e^{i\Phi} \operatorname{sech}(x - X), \quad m^3 = \sigma \tanh(x - X). \qquad (50)$$

Stability of these solutions has a topological origin: domain walls possess a topological charge

$$\sigma = \frac{1}{2} \int_{-\infty}^{\infty} dx \, \partial_x m^3 = \left. \frac{m^3(x)}{2} \right|_{-\infty}^{+\infty} = \pm 1. \qquad (51)$$

Collective coordinates $X$ and $\Phi$ in Eq. (50) represent zero modes associated with the spontaneous breaking of the translational and spin-rotational symmetries by the domain wall. On the two-dimensional manifold

$$-\infty < X < +\infty\,, \quad 0 \le \Phi < 2\pi\,, \tag{52}$$

the symplectic form is

$$F = -2\sigma \mathcal{S}\, dX \wedge d\Phi\,, \tag{53}$$

where $\mathcal{S} = S/a$ is the spin density and $a$ is the atomic lattice period. Both the energy and symplectic form are invariant under infinitesimal translations and rotations generated by vector fields $\partial_X$ and $\partial_\Phi$. By taking the interior products of these vector fields with the symplectic 2-form, we obtain closed 1-forms

$$\iota_{\partial_X} F = -2\sigma \mathcal{S}\, d\Phi \equiv -dP_X\,, \quad \iota_{\partial_\Phi} F = 2\sigma \mathcal{S}\, dX \equiv -dP_\Phi\,, \tag{54}$$

which yield the conserved linear and angular momenta

$$P_X = 2\sigma \mathcal{S}\Phi\,, \quad P_\Phi = -2\sigma \mathcal{S} X\,. \tag{55}$$

Note that $P_X$ is multiple-valued: advancing the azimuthal angle $\Phi$ by $2\pi$ returns the domain wall to the same state, but the linear momentum $P_X$ increases by $4\pi\sigma\mathcal{S}$. This effect is directly tied to the presence of non-contractible loops on a cylinder: for a loop winding $n$ times in the azimuthal direction,

$$\int_C \iota_V F = -\int_C 2\sigma \mathcal{S}\, d\Phi = -4\pi\sigma\mathcal{S} n\,. \tag{56}$$

This violates the necessary condition for the existence of a well-defined conserved momentum $P_V$. See Section 2.3 and Appendix A for details.

The paradox of a multiple-valued linear momentum of ferromagnetic solitons was noted and resolved by Haldane [15]. The field theory of magnetization $\mathbf{m}(x)$ is only a continuum approximation to a lattice theory of spins. In a ferromagnetic chain with spin density $\mathcal{S} = S/a$, translational symmetry is discrete. The elementary translation operator is $e^{iP_X a}$. Shifting the momentum by $4\pi\sigma\mathcal{S}$ multiplies the translation operator by $e^{4\pi i\sigma S}$, which equals 1 for integer and half-integer values of spin length $S$. Thus the multi-valued nature of conserved momentum $P_X$ has no physical consequences.

Alternatively, we may obtain each conserved momentum as the canonical momentum for a suitably symmetric gauge 1-form $A$. The gauge potential

$$A = -2\sigma \mathcal{S} X\, d\Phi \tag{57}$$

is symmetric under global spin rotations $\partial_\Phi$. Hence the conserved angular momentum

$$P_\Phi = \iota_{\partial_\Phi} A = -2\sigma \mathcal{S} X\,. \tag{58}$$

An alternative choice for the gauge potential,

$$A = 2\sigma \mathcal{S}\Phi\, dX\,, \tag{59}$$

is symmetric under translations. It yields the linear momentum

$$P_X = \iota_{\partial_X} A = 2\sigma \mathcal{S}\Phi\,. \tag{60}$$

## 3.2 Vortex in a ferromagnetic film

Consider a ferromagnetic thin film with easy-plane anisotropy, whose energy functional is

$$U[\mathbf{m}(x,y)] = \int dx\,dy\,\frac{1}{2}\left[\partial_x\mathbf{m}\cdot\partial_x\mathbf{m} + \partial_y\mathbf{m}\cdot\partial_y\mathbf{m} + (\mathbf{m}\cdot\mathbf{e}_3)^2\right]. \tag{61}$$

Aside from absolute ground states with uniform $\mathbf{m}(x,y)$ in the easy plane, the energy has local minima in the form of vortices, $\mathbf{m}(x,y) = \mathbf{m}_v(x-X, y-Y)$, where

$$\mathbf{m}_v(0,0) = \pm\mathbf{e}_3, \quad \mathbf{m}_v(x,y) \sim \frac{(x,y,0)}{\sqrt{x^2+y^2}} \text{ as } \sqrt{x^2+y^2} \to \infty. \tag{62}$$

Coordinates of the vortex core $X$ and $Y$ are zero modes associated with translational symmetry. In the 2-dimensional space $(X,Y)$, the symplectic form is

$$F = -4\pi Q\mathcal{S}\,dX \wedge dY, \tag{63}$$

where $Q = \pm 1/2$ is the skyrmion number defined below in Eq. (69) and $\mathcal{S}$ is the spin density. The symplectic form is invariant under translations $\partial_X$ and $\partial_Y$.

Conserved momenta can be defined through the introduction of closed 1-forms

$$\iota_{\partial_X}F = -4\pi Q\mathcal{S}dY \equiv -dP_X, \quad \iota_{\partial_Y}F = 4\pi Q\mathcal{S}dX \equiv -dP_Y, \tag{64}$$

whence the conserved linear momenta

$$P_X = 4\pi Q\mathcal{S}Y, \quad P_Y = -4\pi Q\mathcal{S}X. \tag{65}$$

Alternatively, they can be obtained through the canonical route. The symplectic form can be obtained from a gauge 1-form, $F = dA$. The gauge potentials

$$A = -4\pi Q\mathcal{S}X\,dY, \quad A = 4\pi Q\mathcal{S}Y\,dX, \quad A = -2\pi Q\mathcal{S}(X\,dY - Y\,dX), \tag{66}$$

are invariant under Cartan symmetries $V = \partial_X$, $\partial_Y$, and $X\partial_Y - Y\partial_X$, respectively. By taking the respective inner product $\iota_V A$, we obtain the two conserved linear momenta (65) and the conserved (orbital) angular momentum

$$P_{X\partial_Y - Y\partial_X} = -2\pi Q\mathcal{S}(X^2 + Y^2). \tag{67}$$

# 4 Conserved momenta of the Belavin-Polyakov skyrmion

## 4.1 Heisenberg model in 2 spatial dimensions

In this section, we illustrate the general approach by deriving conserved momenta of a skyrmion in the Heisenberg model in 2 spatial dimensions. In this case, the system has its energy

$$U = \int dx\,dy\,\frac{1}{2}(\partial_x\mathbf{m}\cdot\partial_x\mathbf{m} + \partial_y\mathbf{m}\cdot\partial_y\mathbf{m}). \tag{68}$$

At spatial infinity $\mathbf{m} = (m^1, m^2, m^3)$ must approach a constant value, so that the energy (68) does not diverge. Therefore, we may one-point compactify the infinite plane to a sphere $S^2$.

Belavin and Polyakov [13] showed that this model has a large set of locally stable energy minima whose energy $E = 4\pi|Q|$ is determined by the (integer) skyrmion number

$$Q = \frac{1}{4\pi}\int dx\,dy\,\mathbf{m}\cdot(\partial_x\mathbf{m}\times\partial_y\mathbf{m}). \tag{69}$$

The skyrmion number can be understood as the degree of the configuration map $\mathbf{m} : S^2 \rightarrow S^2$ after the one-point compactification. These solutions can be written down most simply with the aid of the stereographic mapping from the complex plane $\psi$ to the unit sphere $\mathbf{m}$,

$$\psi = \frac{m^1 + i m^2}{1 + m^3}, \quad \bar{\psi} = \frac{m^1 - i m^2}{1 + m^3}. \tag{70}$$

We essentially identified both the domain and the target of our configuration map $\mathbf{m}$ as the Riemann sphere. Any solution with $Q > 0$ can be expressed as a meromorphic function $z \mapsto \psi(z)$. But meromorphic functions defined on the whole Riemann sphere are rational, $\psi(z) = P(z)/R(z)$, and the topological degree of $\psi$ is $\max\{\deg P, \deg R\}$, where deg is the degree of $P$ and $R$ as polynomials. Therefore, the collection of $Q = 1$ configurations, or skyrmions, are identified by the Möbius transformations

$$\psi(z) = \frac{Az + B}{Cz + D}. \tag{71}$$

## 4.2 Belavin-Polyakov skyrmion

We consider a skyrmion in the background of the vacuum $\mathbf{m} = (0, 0, -1)$. This requires $\psi(z) \rightarrow \infty$ as $z \rightarrow \infty$. These skyrmion states can be parametrized in terms of two complex numbers $Z$ and $\zeta$:

$$\psi(z) = \frac{z - Z}{\zeta}. \tag{72}$$

Complex coordinate $Z = X + iY$ encodes the location of the skyrmion center $(X, Y)$, where $\psi = 0$ and thus $\mathbf{m} = (0, 0, 1)$. Parameter $\zeta$ serves a dual purpose. Its magnitude $\rho = |\zeta|$ determines the radius of the skyrmion defined as the distance from the center at which $m^3 = 0$. The phase $\alpha = \arg \zeta$ serves as the global rotation angle for magnetization in the $(m^1, m^2)$ plane.

The 4 real parameters $X$, $Y$, $\alpha$, and $\rho$ can be viewed as collective coordinates of the skyrmion's zero modes because the energy $E = 4\pi$ does not depend on them. The first two of the zero modes are translations, the third is a spin rotation, the global symmetries of the Heisenberg model. The zero mode associated with $\rho$ is the dilation, which also preserves the energy of the Heisenberg model in $d = 2$:

$$\mathcal{L}_V U = 0 \quad \text{for } V = \partial_X, \partial_Y, \partial_\alpha, \partial_\rho. \tag{73}$$

## 4.3 Symplectic form

We next obtain the symplectic form $F$ (47) by computing the gyroscopic matrix $F_{ij}$ (48).

Because $\mathbf{m}$, like $\psi$, depends on $X$ and $Y$ through the differences $x - X$ and $y - Y$, the partial derivatives with respect to $X$ and $Y$ can be replaced with the gradients, $\partial_X \mathbf{m} = -\partial_x \mathbf{m}$, $\partial_Y \mathbf{m} = -\partial_y \mathbf{m}$. Therefore the gyroscopic coefficient $F_{XY}$ is proportional to the skyrmion number (69). We obtain

$$F_{XY} = -F_{YX} = -4\pi\mathcal{S}, \tag{74}$$

where $\mathcal{S}$ is the spin density.

The gyroscopic coefficient $F_{\alpha\rho}$ is divergent in an infinite plane. Limiting the magnet to a disk of large radius $R$ centered at the origin yields the following asymptotic result:

$$F_{\alpha\rho} = -F_{\rho\alpha} \sim -4\pi\mathcal{S}\rho\left(2\ln\frac{R}{\rho} - 1\right), \quad R \gg X, Y, \rho. \tag{75}$$

The rest of the gyroscopic coefficients vanish. We thus arrive at the symplectic form

$$F = -4\pi\mathcal{S}\, dX \wedge dY - 4\pi\mathcal{S}\rho\left(2\ln\frac{R}{\rho} - 1\right) d\alpha \wedge d\rho. \tag{76}$$

Translations and global spin rotations preserve the symplectic form:

$$\mathcal{L}_V F = 0 \quad \text{for } V = \partial_X, \, \partial_Y, \, \partial_\alpha. \tag{77}$$

However, dilations do not. With the aid of Cartan's lemma, we find

$$\mathcal{L}_{\partial_\rho} F = \iota_{\partial_\rho} dF + d \, \iota_{\partial_\rho} F = \partial_\rho F_{\alpha\rho} \, d\alpha \wedge d\rho \neq 0. \tag{78}$$

Although dilations preserve the potential energy of a skyrmion, they change the symplectic form. It is then not possible to construct a conserved momentum associated with the dilation vector field $\partial_\rho$. The resulting 1-form

$$\iota_{\partial_\rho} F = 4\pi \mathcal{S} \rho \left( 2 \ln \frac{R}{\rho} - 1 \right) d\alpha \tag{79}$$

is not closed and therefore cannot be expressed as the external derivative of a 0-form (conserved momentum).

A formal way out is to use the vector field

$$V = \frac{1}{4\pi\rho \left( 2 \ln \frac{R}{\rho} - 1 \right)} \frac{\partial}{\partial \rho} = \frac{\partial}{\partial \Sigma}, \tag{80}$$

where we reparametrized the skyrmion radius $\rho$ in terms of its "area"

$$\Sigma = 4\pi\rho^2 \ln \frac{R}{\rho}. \tag{81}$$

This variable is more natural as the symplectic form simplifies to

$$F = -4\pi \mathcal{S} \, dX \wedge dY - \mathcal{S} \, d\alpha \wedge d\Sigma. \tag{82}$$

Both the symplectic form and the energy are invariant under the area expansions $\partial_\Sigma$.

## 4.4 Conserved momenta

The four Cartan symmetries $\partial_X$, $\partial_Y$, $\partial_\alpha$, and $\partial_\Sigma$ give rise to the four conserved momenta:

$$P_X = 4\pi \mathcal{S} Y, \quad P_Y = -4\pi \mathcal{S} X, \quad P_\alpha = \mathcal{S} \Sigma, \quad P_\Sigma = -\mathcal{S} \alpha. \tag{83}$$

The first two of these have been derived previously [5, 16, 17].

Conserved momentum $P_\alpha$ is associated with global spin rotations $\partial_\alpha$ that preserve the background vacuum. It is reasonable to suspect that this quantity is merely the total spin of the skyrmion $S^3$. Indeed, it straightforward to check that the spin of the skyrmion relative to that of the $\mathbf{m} = (0, 0, -1)$ vacuum is

$$S^3 = \mathcal{S} \int_{r<R} d^2 r \, (m^3 + 1) = \mathcal{S} \Sigma. \tag{84}$$

Its divergence, and the need for a long-distance cutoff $R$, is associated with a slow return of $\mathbf{m}$ to its asymptotic value at spatial infinity in the scale-free Heisenberg model.

Lastly, the symmetry $\partial_\Sigma$ of expanding the skyrmion's effective area, which preserves both the energy $U$ and symplectic form $F$, gives rise to conserved momentum $P_\Sigma = -\mathcal{S} \alpha$ proportional to the global spin angle $\alpha$.

It is worth noting that momentum $P_\Sigma$ is multi-valued just like the linear momentum $P_X$ of the domain wall. Again, this is associated with nontrivial topology of the collective-coordinate manifold $q = \{X, Y, \alpha, \Sigma\}$ that has non-contractible loops winding in the $\alpha$ direction. The multi-valued nature of this conserved momentum points to a discrete nature of the coordinate $\Sigma$. Indeed, the spin of the skyrmion (84) has a quantum nature and can only be incremented in integer steps.

## 4.5   Low-energy dynamics of a skyrmion

In the absence of perturbations, the four conserved momenta (83) of a BP skyrmion remain constant. The position, size, and global azimuthal spin angle do not change with time. Weak perturbations will induce slow motion of these collective variables. This low-energy dynamics can be captured by an effective theory with four collective coordinates $q = \{X, Y, \alpha, \Sigma\}$, whose time evolution is governed by Eq. (2). The gyroscopic tensor has nonzero components

$$F_{XY} = -F_{YX} = -4\pi\mathcal{S}, \quad F_{\alpha\Sigma} = -F_{\Sigma\alpha} = -\mathcal{S}. \tag{85}$$

Translational motion of ferromagnetic solitons, parametrized by the pair of coordinates $(X, Y)$, is well understood [1, 2, 16], so we focus on the other two coordinates, $\alpha$ and $\Sigma$.

### 4.5.1   Magnetic field

A practical example of an external perturbation is an applied magnetic field $\mathbf{h} = (0, 0, h)$ parallel to magnetization of the uniform ground state $\mathbf{m} = (0, 0, -1)$. It couples directly to the skyrmion spin (84) via a Zeeman term in the energy, $U = -\gamma h \mathcal{S} \Sigma$, where $\gamma$ is the gyromagnetic ratio. This perturbation induces uniform precession of the global azimuthal angle at the Larmor frequency, $\dot{\alpha} = -\gamma h$, while the position $(X, Y)$ and spin $\mathcal{S}\Sigma$ of the skyrmion remain static.

### 4.5.2   Spin-polarized current

Another readily accessible perturbation is a spin-polarized electric current injected into the center of a skyrmion and flowing outward. Through the mechanism of adiabatic spin transfer, conduction electrons passing through an area of inhomogeneous magnetization exchange angular momentum with the ferromagnet, thereby applying what's known as the adiabatic spin-transfer torque to the magnetization [18, 19]. As one of us discussed elsewhere [8], the adiabatic spin-transfer torque can be viewed as an instance of the gyroscopic force if we expand the dynamical system to include the electric charge $Q$ of the injected current $I = \dot{Q}$. The current creates a gyroscopic force $F_{\alpha Q}\dot{Q}$ for the $\alpha$ coordinate, with the gyroscopic coefficient $F_{\alpha Q} = \mathcal{P}\hbar/e$, where $e$ is the electron charge and $\mathcal{P}$ is the degree of spin polarization in the current. In an inverse effect, a time-dependent magnetization induces an electric field [20] with Cartesian components

$$E_n = \frac{\mathcal{P}\hbar}{2e}\mathbf{m} \cdot (\partial_n \mathbf{m} \times \dot{\mathbf{m}}), \tag{86}$$

known as a "spin electromotive force" [21, 22]. Precession of the global azimuthal angle $\alpha$ of the BP skyrmion creates a voltage $F_{Q\alpha}\dot{\alpha}$ between the center of the skyrmion and infinity, where $F_{Q\alpha} = -F_{\alpha Q} = -\mathcal{P}\hbar/e$. Equation of motion (2) for collective coordinate $\alpha$ then reads

$$-\mathcal{S}\dot{\Sigma} + \mathcal{P}\hbar\dot{Q}/e = 0. \tag{87}$$

The skyrmion area $\Sigma$ expands or contracts, depending on the direction of the spin-polarized current. This result can also be obtained from conservation of angular momentum: the spin of an electron injected at the center of the skyrmion and flowing outward changes its component $S^3$ from $+\hbar/2$ to $-\hbar/2$ as it adiabatically follows the local direction of magnetization; each injected electron thus adds angular momentum $S^3 = +\hbar$ to the skyrmion.

# 5 Conclusion

We have shown that Cartan's theory of differential forms is a natural mathematical language for the dynamics of collective coordinates $q = \{q^1, \ldots q^N\}$ of a ferromagnetic soliton. The gyroscopic tensor $F_{ij}$ [3,4] gives rise to a closed 2-form $F = \frac{1}{2} F_{ij}\, dq^i \wedge dq^j$. The equations of motion can be written in a coordinate-free form, $\iota_D F = -dU$, where $U(q)$ is the potential energy and $D = \dot{q}^i \partial_i$ is the convective derivative.

A continuous symmetry of the soliton, parametrized by vector fields $V = V^i \partial_i$, is associated with a closed 1-form $\iota_V F$, which can be expressed in terms of its potential, a 0-form: $\iota_V F = -dP_V$. This potential $P_V$ is the conserved momentum for symmetry $V$.

Conserved momenta are not always well defined. They can be multivalued if the manifold of collective coordinates has loops not contractible to a point. Conserved momentum $P_V$ can be increased by $\Delta P_V = -\int_C \iota_V F$ by going around such a loop.

We used this formalism to obtain the full set of conserved momenta for the Belavin-Polyakov skyrmion in the two-dimensional Heisenberg model. Its zero modes include two translations, a global spin rotation, and dilation. We obtained the vector field $\partial_\Sigma$ (80) of the dilation symmetry and showed that the corresponding conserved momentum is proportional to the angle $\alpha$ of global spin rotation. The four conserved momenta of the Belavin-Polyakov skyrmion are listed in Eq. (83). They represent low-energy degrees of freedom of this soliton. Whereas the dynamics of the center of a skyrmion is already well understood, we presented the dynamics of the global azimuthal angle and of the skyrmion area (spin) under applied magnetic field or spin-polarized current.

## Acknowledgments

We thank Sayak Dasgupta, Connor Grady, Boris Ivanov, and Se Kwon Kim for helpful discussions. This research has been supported in part by the US NSF Grants PHY-1607611 (Aspen Center for Physics) and PHY-1748958 (Kavli Institute for Theoretical Physics).

## A De Rham's theorem

De Rham's theorem [10] gives a necessary and sufficient condition under which a closed $n$-form $\omega$ has a potential $\eta$,

$$\omega = d\eta, \tag{A.1}$$

globally defined on a manifold $\mathcal{M}$.

For every $n$-cycle[2] $C$ of $\mathcal{M}$, we can define the *period*

$$\text{per}\, C = \int_C \omega. \tag{A.2}$$

A consequence of de Rham's theorem is that $\omega$ has a globally defined potential if every $n$-cycle $C$ in $\mathcal{M}$ has a vanishing period.

The period (A.2) is a topological quantity: a continuous deformation of the $n$-cycle leaves the period unchanged. Roughly speaking, homotopic $n$-cycles belong to the same homology class, whose members have the same period (A.2).

---

[2] By an $n$-cycle we mean a compact $n$-dimensional submanifold without boundary [10]. This is a slight abuse of notation: it captures the idea of actual $n$-cycles in homology theory, but the formal definition involves unnecessary complication.

In a manifold with simple topology (such as a Euclidean space), every $n$-cycle can be continuously contracted to a point, so $\int_C \omega = 0$ always. The global definition of the potential is ensured.

A manifold with more complex topology may have non-contractible $n$-cycles. If the period does not vanish for any class of non-contractible $n$-cycles, the potential can be defined locally but not globally.

A well-known example is the magnetic field of a magnetic monopole of strength $g$, given by the 2-form

$$F = \frac{g}{r^3}(x\,dy \wedge dz + y\,dz \wedge dx + z\,dx \wedge dy), \quad r = \sqrt{x^2 + y^2 + z^2}. \tag{A.3}$$

This 2-form is well defined and closed, $dF = 0$, on $\mathcal{M} = \mathbb{R}^3 \setminus (0,0,0)$, the 3-dimensional Euclidean space with the origin excluded. This manifold has non-contractible 2-cycles, e.g., a 2-sphere $S$ enclosing the origin. Any such surface sees a magnetic flux

$$\mathrm{per}\, S = \int_S F = 4\pi g \neq 0. \tag{A.4}$$

Therefore a gauge potential $A$ such that $F = dA$ cannot be defined globally on $\mathcal{M}$.

On the other hand, if we exclude not just the origin but an entire line $L$ leading from the origin to infinity, such as

$$x = y = 0, \quad z \geq 0, \tag{A.5}$$

the resulting manifold $\mathcal{M}' = \mathbb{R}^3 \setminus L$ will have no non-contractible 2-cycles. Then the period vanishes for any 2-cycle and a gauge potential can be globally defined on $\mathcal{M}'$, e.g.,

$$A = \frac{g}{r(r-z)}(y\,dx - x\,dy). \tag{A.6}$$

In physical terms, the excluded line (A.5) carries a return magnetic flux $-4\pi g$ and is known as the Dirac string.

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
