# Peer review of "Conserved momenta of ferromagnetic solitons through the prism of differential geometry"

_SciPost Physics, doi:SciPost Phys. 11, 108 (2021)_

## Round 1 · Referee Report · Anonymous (Referee 2) · 2021-6-22

Report

In this manuscript, the authors have revisited the long-standing problem of conserved angular momenta of a ferromagnetic system. They proposed a differential form based on the Cartan's theory to describe the dynamics of collective coordinates. The manuscript is well written I found it interesting its result interesting. I think the manuscript satisfies the journal's criteria and I support publication of this manuscript in SciPost Physics.
I have only one optional comment: I think it is possible to extend the current work to spin torsion tensor formalism, discussed in the gravity theories. I am wondering if the problem of definition of proper momenta in the magneto-elastic media can also be resolved with this approach. If yes, it should be instructive for readers to point it out in the manuscript.
  • validity: top
  • significance: good
  • originality: high
  • clarity: high
  • formatting: excellent
  • grammar: excellent

Author:  Oleg Tchernyshyov  on 2021-07-01  [id 1540]

(in reply to Report 2 on 2021-06-22)

We thank the referee for a positive assessment of the manuscript. The specific comments are addressed below.

I think it is possible to extend the current work to spin torsion tensor formalism, discussed in the gravity theories.

We are not sure that we understand what the referee suggests. The spin torsion tensor is the antisymmetric part of the Levi-Civita connection that is assumed to vanish in Einstein's theory of gravity. It is considered in extensions of general relativity and may be relevant to systems with macroscopic amounts of spin (like a ferromagnet). However, here we are concerned with the dynamics of ferromagnetic solitons. It is not clear to us whether consideration of torsion at the level of general relativity will give rise to qualitatively new behavior of a ferromagnet such as the appearance of a new type of magnetic force besides conservative and gyroscopic (see Sec. 1.1). Perhaps we misunderstand the comment and the referee meant something entirely different?

I am wondering if the problem of definition of proper momenta in the magneto-elastic media can also be resolved with this approach. If yes, it should be instructive for readers to point it out in the manuscript.

Indeed, one of us has been pondering a related question: can the linear momentum of a ferromagnetic soliton be transferred to the atomic lattice? The angular momentum clearly can be (the Einstein-de Haas effect). However, our preliminary analysis yields a negative answer for the linear momentum, which is a bit of a surprise. We are not ready to give a definitive answer at this time and hope to be able to address it at a later point.

---

## Round 1 · Referee Report · Anonymous (Referee 1) · 2021-6-22

Strengths

  1. The paper is written clearly and methodologically.
  2. The mathematical framework explained is important for many applications in magnetism.
  3. The manuscript provides a better understanding of the Belavin-Polyakov solution in field theory.

Weaknesses

A reader might be interested not only in a mathematical toolbox, but also in a more in-depth insight into physical conclusions.

Report

The authors present a discussion about conservation laws for the topological magnetic solitons and identify a proper mathematical framework to find symmetries of the given texture and derive the respective conserved momenta. This approach is used to analyze the Belavin-Polyakov soliton. The discussed approach is well-timed because of recent interest to the complex magnetic objects, where methods of the differential geometry help to understand the physics. The Belavin-Polyakov solution itself is one of the most known fundamental models in the field theory, and a better understanding of its properties should provide an impact beyond the magnetic community. That’s why, I believe, the manuscript is of interest and can be published in SciPost Physics.

I have some comments, which are listed according to the text in this manuscript.

After Eq. (4) authors say that the gauge potential is not a physical quantity. However, its physical meaning is well discussed in literature. For example, one can mention the Feynman’s question “Is the vector potential a “real” field?”

In a discussion of the multiple-valued of $P_X$ for the domain wall, I wonder about its relation with the Berry phase. One can suggest that such quantities can obtain an additional physical sense in specific cases like motion under the action of spin-orbit torques.

If I understand correctly, the new conserved momenta are related with the finite size of the system due to the cutting radius $R$. This sounds as an additional condition which determines the conservation according to $O(X^n, Y^n)$ with some $n$ in Eq. (75). Then, the presense of these momenta can be expexted to be pronounced in the relaxation dynamics of the soliton, see DOI:10.1103/PhysRevB.90.174428 for example.

There are many works on the so-called dynamic solitons (see also Ref. [1] in the manuscript), stabilization of whose is reached by a steady rotation of the magnetization phase $\phi = \phi_0 + \omega t$. Is the second pair of momenta, $P_\alpha$, and $P_\Sigma$, related to this mechanism?

Minor remarks: - sentence before Eq. (3), it seems like a word is missing: “... of a massless electrically charged [particle] in a...”

Requested changes

Besides misprints, I believe that the manuscript should strongly benefit from expanding the discussion on physical consequences and, especially, new pair of conserved momenta.

  • validity: high
  • significance: high
  • originality: high
  • clarity: high
  • formatting: excellent
  • grammar: excellent

Author:  Oleg Tchernyshyov  on 2021-07-01  [id 1539]

(in reply to Report 1 on 2021-06-22)

We thank the referee for a detailed report and for constructive criticism. Below we address the referee's queries.

A reader might be interested not only in a mathematical toolbox, but also in a more in-depth insight into physical conclusions.

In the newly added Sec. 4.5 we discuss the dynamics of the BP skyrmion under the action of an applied magnetic field and a spin-polarized current injected at the skyrmion center. Whereas translational motion of a skyrmion has been widely discussed, the dynamics of its global azimuthal angle and area (spin) should be of interest to physicists.

After Eq. (4) authors say that the gauge potential is not a physical quantity. However, its physical meaning is well discussed in literature. For example, one can mention the Feynman’s question “Is the vector potential a “real” field?”

The gauge potential itself is not physically measurable and can be changed by gauge transformations. A physically relevant quantity is the circulation of the gauge potential around a loop, or, equivalently, the magnetic flux through the loop. For an infinitesimal loop this line integral gives the strength of the magnetic field and for a large loop the Aharonov-Bohm phase. So, although the gauge potential itself is not a physical quantity, it can be used to extract physically meaningful information.

In a discussion of the multiple-valued of $P_X$ for the domain wall, I wonder about its relation with the Berry phase. One can suggest that such quantities can obtain an additional physical sense in specific cases like motion under the action of spin-orbit torques.

All conserved momenta of a ferromagnetic soliton are intimately related to the Berry phase of precessing spins. The gauge potential term in the Lagrangian (3) expresses the spin Berry phase in the action. As we explain in Sec. 2.4, conserved momenta can be derived directly from the (suitably chosen) gauge potential.

If I understand correctly, the new conserved momenta are related with the finite size of the system due to the cutting radius $R$. This sounds as an additional condition which determines the conservation according to $O(X^n,Y^n)$ with some $n$ in Eq. (75). Then, the presense of these momenta can be expexted to be pronounced in the relaxation dynamics of the soliton, see DOI:10.1103/PhysRevB.90.174428 for example.

We are not exactly sure what the referee implies in regards to Eq. (75). The gyroscopic coefficients are not particularly sensitive to X and Y as long as the skyrmion radius and displacement are small relative to the sample size. Nonetheless, we agree that these conserved momenta will be useful in describing the dynamics of a skyrmion in the presence of viscous forces (Gilbert's damping) purposefully omitted in this paper. In one of our earlier papers, doi:10.1103/PhysRevB.95.180408, we used precisely this strategy to describe the annihilation of two domain walls in a one-dimensional wire.

There are many works on the so-called dynamic solitons (see also Ref. [1] in the manuscript), stabilization of whose is reached by a steady rotation of the magnetization phase $\phi = \phi_0 + \omega t$. Is the second pair of momenta, $P_\alpha$ and $P_\Sigma$, related to this mechanism?

The referee is right. In the newly added Sec. 4.5 we demonstrate that the azimuthal angle of a BP skyrmion uniformly precesses in an applied magnetic field.

sentence before Eq. (3), it seems like a word is missing: “... of a massless electrically charged [particle] in a...”

This typo has been fixed.

---

## Round 2 · Referee Report · Anonymous (Referee 2) · 2021-10-13

Report
The present manuscript addresses a challenge in the relation between symmetries and conservation laws in ferromagnets.
In FMs, since there is a background gauge field it is challenging to make a gauge-invariant energy-momentum tensor and apply Noether's theorem. Even though the main problem which has been considered here has been known for many years in the community of magnetism, thanks to the recent advances in spintronics, which make manipulation and interaction of nonlinear magnetic solitons and magnetic excitations possible, there is a new interest in this topic. For example these recent papers: PHYSICAL REVIEW B 88, 144413 (2013) and PHYSICAL REVIEW B 98, 224401 (2018).
Here, in this paper, the authors have developed a new approach (especially in the context of magnetism) based on Cartan's theory which resolves part of gauge-invariant difficulties, and related issues such as the relation of canonical and conserved momenta, and also possible path dependence of the conserved momenta in the collective coordinates. In addition, the rigorousness and power of Cartan's differential geometry, make the formalism developed in this paper a suitable framework for future investigations about ferromagnetic solitons. So overall, the paper has developed an approach that resolves theoretical difficulties in a more rigorous and less subtle manner. Therefore, it is a timely study in this topic that can serve as a part of theoretical developments and I suggest its publication in SciPost Physics.
In FMs, since there is a background gauge field it is challenging to make a gauge-invariant energy-momentum tensor and apply Noether's theorem. Even though the main problem which has been considered here has been known for many years in the community of magnetism, thanks to the recent advances in spintronics, which make manipulation and interaction of nonlinear magnetic solitons and magnetic excitations possible, there is a new interest in this topic. For example these recent papers: PHYSICAL REVIEW B 88, 144413 (2013) and PHYSICAL REVIEW B 98, 224401 (2018).
Here, in this paper, the authors have developed a new approach (especially in the context of magnetism) based on Cartan's theory which resolves part of gauge-invariant difficulties, and related issues such as the relation of canonical and conserved momenta, and also possible path dependence of the conserved momenta in the collective coordinates. In addition, the rigorousness and power of Cartan's differential geometry, make the formalism developed in this paper a suitable framework for future investigations about ferromagnetic solitons. So overall, the paper has developed an approach that resolves theoretical difficulties in a more rigorous and less subtle manner. Therefore, it is a timely study in this topic that can serve as a part of theoretical developments and I suggest its publication in SciPost Physics.

---

## Round 2 · Author Response

Dear Editor,
We are resubmitting our manuscript in response to the referee reports.
The reports are overall positive, with both reviewers recommending publication of the manuscript in SciPost Physics. The referees raised some questions and suggested a number of improvements, for which we are grateful. We have expanded the manuscript in accordance with these suggestions. Below we outline the changes in the manuscript. Specific queries of the referees have been addressed in our replies to them.
Referee 1 wrote: "the manuscript should strongly benefit from expanding the discussion on physical consequences and, especially, new pair of conserved momenta." We have added Subsection 4.5 "Low-energy dynamics of a skyrmion," in which we spell out the physical significance of the new pair of conserved momenta---the global azimuthal angle and skyrmion spin. In the presence of perturbations beyond the ideal Heisenberg model, these two quantities evolve in time in a prescribed manner. The application of a uniform magnetic field parallel to the ground-state magnetization induces a steady precession of the global azimuthal angle, whereas the injection of a spin-polarized electric current at the center of the skyrmion leads to an expansion or contraction of the skyrmion area (spin). We have also added a few sentences mentioning these consequences to Section 5 "Conclusion."
We hope that you will find the revised manuscript suitable for publication and look forward to your reply.
With best regards,
Xingjian Di
Oleg Tchernyshyov
We are resubmitting our manuscript in response to the referee reports.
The reports are overall positive, with both reviewers recommending publication of the manuscript in SciPost Physics. The referees raised some questions and suggested a number of improvements, for which we are grateful. We have expanded the manuscript in accordance with these suggestions. Below we outline the changes in the manuscript. Specific queries of the referees have been addressed in our replies to them.
Referee 1 wrote: "the manuscript should strongly benefit from expanding the discussion on physical consequences and, especially, new pair of conserved momenta." We have added Subsection 4.5 "Low-energy dynamics of a skyrmion," in which we spell out the physical significance of the new pair of conserved momenta---the global azimuthal angle and skyrmion spin. In the presence of perturbations beyond the ideal Heisenberg model, these two quantities evolve in time in a prescribed manner. The application of a uniform magnetic field parallel to the ground-state magnetization induces a steady precession of the global azimuthal angle, whereas the injection of a spin-polarized electric current at the center of the skyrmion leads to an expansion or contraction of the skyrmion area (spin). We have also added a few sentences mentioning these consequences to Section 5 "Conclusion."
We hope that you will find the revised manuscript suitable for publication and look forward to your reply.
With best regards,
Xingjian Di
Oleg Tchernyshyov

---

## Editorial Decision

published